# Low-Illumination Image Enhancement Using Local Gradient Relative Deviation for Retinex Models

**Biao Yang** [1,2,3], **Liangliang Zheng** [1,2,3,*], **Xiaobin Wu** [1,2,3], **Tan Gao** [1,2,3] **and Xiaolong Chen** [1,2,3]

1 Changchun Institute of Optics, Fine Mechanics and Physics, Chinese Academy of Sciences, Changchun 130033, China; yangbiao211@mails.ucas.ac.cn (B.Y.); wuxiaobin20@mails.ucas.ac.cn (X.W.); gaotan19@mails.ucas.ac.cn (T.G.); chenxiaolong20@mails.ucas.ac.cn (X.C.)
2 University of Chinese Academy of Sciences, Beijing 100039, China
3 Key Laboratory of Space-Based Dynamic & Rapid Optical Imaging Technology, Chinese Academy of Sciences, Changchun 130033, China
* Correspondence: zhengliangliang@ciomp.ac.cn

**Abstract:** In order to obtain high-quality images, the application of low-illumination image enhancement techniques plays a vital role in enhancing the overall visual appeal. However, it is particularly difficult to enhance an image while maintaining the original information of the scene. The augmentation method based on Retinex theory is widely considered as one of the representative techniques for such problems, but this method still has some limitations. First of all, noise is easily ignored in the process of model building, and the robustness of the model needs to be improved. Secondly, the image decomposition is less effective, so that part of the image information is not effectively presented. Finally, the optimization procedure is computationally complicated. This paper introduces a novel approach for enhancing low-illumination images by utilizing the relative deviation of local gradients. The proposed method aims to address the challenges associated with low-illumination images and offers a solution to these issues. In this paper, local gradient relative deviation is used as a constraint term and a noise term is added to highlight the image texture and structure and improve the robustness of the models, considering that $L_P$ achieves piecewise smoothing with better sparsity compared to the sum norm commonly used by $L_1$ and $L_2$ norms. In this paper, the $L_2 - L_P$ norm is used to constrain the model, which smooths the illumination component and better preserves the details of the reflectance component. In addition, to efficiently solve the optimization problem, the alternating direction multiplier method is chosen to transform the optimization process into the solution of several sub-problems. In comparison to traditional Retinex models, the proposed method excels in its ability to simultaneously enhance the image and suppress noise effectively. The experimental outcomes demonstrate the effectiveness of the proposed model in enhancing both simulated and real data. This approach can be applied to low-illumination remote sensing images to obtain high-quality remote sensing image data.

**Keywords:** low-illumination image; local gradient relative deviation; $L_P$ constraint; Retinex model





## 1. Introduction

Image quality is affected by numerous factors, both in everyday life and in space remote sensing imaging. These factors encompass the light intensity of the imaging device as well as the imaging environment itself. For this purpose, low illumination enhancement techniques have been widely used to improve the quality of captured images. Recently, several types of methods have been proposed by scholars to deal with dark images, including histogram equalization, Retinex decomposition, and deep learning, among others. Histogram equalization uses a histogram to count the gray level distribution of the image, show the gray level of each pixel in the image in the form of occurrence frequency or number, and evenly distribute the gray level of the image with dense distribution, so as to improve the image contrast and information. Kim et al. [1] proposed adaptive histogram

equalization (AHE), an algorithm with higher complexity. In addition, Reza et al. [2] presented an algorithm using contrast-limited adaptive histogram equalization (CLAHE), which solves the problem of noise and excessive contrast enhancement by limiting the height of each sub-block histogram. Kang et al. [3] proposed the adaptive height modified histogram equalization algorithm (AHMHE), which aims to establish the mapping function of the adaptive height correction histogram and enhance the local contrast by combining the relationship between adjacent pixels. Chang et al. [4] introduced a technique known as automatic contrast finite adaptive histogram equalization with double gamma correction, which reassigns the histogram of CLAHE [2] blocks according to the dynamic range of each block, and performs double gamma correction to enhance brightness. This approach is more suitable for image enhancement in dark regions and can reduce artifacts due to over-enhancement. Pankaj Kandhway et al. [5] introduced a novel sub-histogram equalization method based on adaptive thresholds, which can enhance contrast and brightness while preserving basic image features.

In Retinex theory, the image is obtained via the combined action of the illuminance component and the object reflectance component ($S = R \circ L$, which references the observed image $S$, illuminance component $L$, and reflectance component $R$, where $\circ$ represents element multiplication). The image enhancement algorithm rooted in the Retinex theory involves decomposing the image into two distinct components, the object reflection component and the illumination component. then applying a gamma function to the illumination component, and, finally, combining the two layers to obtain the enhancement result. Jobson et al. [6] proposed the center/surround model using a Gaussian low-pass filter and a logarithmic method. The model included a single-scale Retinex model (SSR) and a multi-scale color restoration Retinex model (MSRCR). The Retinex theory addresses the problem of image separation into object reflectance and illumination components. However, this problem is usually undetermined. Therefore, the variational model was proposed by Kimmel et al. [7]. There is a spatial correlation between the object reflectance and the illuminance component, which can have an effect on the object reflectance during image decomposition. Fu et al. believed that both of them should be estimated at the same time [8]. Based on this, Fu et al. [9] introduced the weighted variational model.

For images with uneven lighting, Wang et al. [10] presented the natural retention enhancement algorithm. Based on the traditional Retinex theory, Guo et al. [11] proposed an image enhancement method by estimating an illuminance map. In this method, the goal is to identify the highest value in the RGB channel and create an illumination map by extracting the maximum grayscale value from the color image across the three channels. By rectifying the initial illumination map using this technique, an improved image can be achieved. Dong et al. [12] proposed a low-illumination enhancement algorithm based on dark channel de-fogging. This method enhanced the image by using de-fogging processing method by taking advantage of the grayscale intensity of a low illumination image after inversion and the approximate gray value of image with fog. Hao et al. [13] proposed an improved model based on the semi-decoupling decomposition (SDD) method. In this model, the decomposition is completed by using the pseudo-decoupling mode. The illuminance component is calculated based on the Gaussian variation of the image, while the reflectivity is calculated jointly from the input image and the illuminance component. Cai et al. [14] proposed the Joint Internal and External Prior (JieP) model based on Retinex. However, the model tends to over-smooth the illumination and reflectivity components of the scene. Li et al. [15] proposed a Retinex enhancement method based on robust structure. The objective of the Retinex theory of robust structure is to enhance the effectiveness of algorithms for improving low-illumination images by considering noisy maps.

Ren et al. [16] proposed the joint enhancement and denoising model based on sequence decomposition (JED). The Retinex model is used to sequentially decompose the images to obtain uniform illumination components and noiseless reflectance components. Xu et al. [17] presented a model of structure and texture perceptual reconstruction (STAR) for enhancing low illumination images. The approach of STAR incorporates an exponential

filter that is specifically designed to extract accurate structure and texture from the image, utilizing specific parameters. The plug-and-play Retinex low-light enhancement model, proposed by Lin in 2022 [18], takes a non-convex $L_P$ constraint and applies a contractive mapping to the illumination layer.

The development of deep learning techniques has spawned a wide range of techniques specifically designed to enhance low-illumination images. The LLNet [19] method uses a dark image with added noise and enhanced image pairs as training. Method [20] is an unsupervised Gan network method. The network of this method is trained in the absence of image pairs. Global–local processing refers to the approach used to handle various illumination conditions present in the input image. Both Retinex network [21] and DeepUPE [22] embrace the principles of the Retinex theory and utilize them as the foundation for their network architectures. Nevertheless, the deep learning approaches necessitate an extensive volume of training data, and the illumination layer can generate artifacts when confronted with test images exhibiting distinct characteristics compared to the training data.

This manuscript introduces an innovative optimization model for Retinex. In this model, from the perspective of texture and structure, the local gradient relative deviation is used as a constraint term in the horizontal and vertical directions, to, respectively, highlight texture and structure. In addition, we incorporate a noise component into the model to enhance its robustness. The features of ambient illumination and object reflectivity are considered in this paper. Ideally, the illumination should be overall smooth, with more detail in the object's reflectivity. When compared to the widely employed $L_1$ and $L_2$ norms, the $L_p$ norm exhibits superior sparsity characteristics, particularly for piecewise smoothing. Therefore, $L_2 - L_P$ norm is adopted in this paper to constrain the local gradient deviation of the object reflectance and the local gradient deviation of the illuminance component, respectively, to better retain the reflectance information while smoothing the illuminance component. Innovations in the research content of this paper come from the literature we have read. According to the analytical quantitative evaluation, the performance of the proposed method surpasses that of the aforementioned methods. The proposed technique has shown promising results in remote sensing. The following list outlines the various innovations presented in this paper:

1.  In this paper, from the perspective of texture and structure, we highlight texture and structure by using local gradient relative deviations in horizontal and vertical directions, respectively, as constraint terms. Furthermore, to enhance the robustness of the model, we introduce a noise term into the equation;
2.  This paper takes into account the attributes of the illumination component as well as the reflectance properties of the object, and uses the $L_2 - L_P$ norm for constraint to smooth the illumination component while better preserving the details of the reflectance component.
3.  The experimental results demonstrate that the proposed method exhibits excellent stability and convergence, making it particularly beneficial for remote sensing images.

The remaining sections of the paper are outlined below. Section 2 of the paper is devoted to the exposition of Retinex theory and its closely related analysis. Section 3 presents a comprehensive exposition of the methodological principles employed in this paper. Section 4 describes the methodology and experimental results of the subjective and objective analyses. Section 5 is dedicated to the presentation of the conclusions of the paper, summarizing the main findings and outcomes of the study.

## 2. Related Work

### 2.1. Retinex Model

The Retinex theory models the color perception of the human visual system, aiming to decompose the observed image $S \in R^{n*m}$ into illuminance component and object reflectance component, namely,

$$S = R \circ L \tag{1}$$

In the formula, $L \in R^{n*m}$ represents the brightness of the object, namely, the scene illuminance component. The physical attributes of the object, specifically, the reflectance component, are represented by $R \in R^{n*m}$; $\circ$ represents element multiplication. The illumination component $L \in R^{n*m}$ and the reflectance $R \in R^{n*m}$ can be estimated alternatively by the following formula:

$$L = S \oslash R, R = S \oslash L \tag{2}$$

where $\oslash$ represents element division. In practice, we use $L = S \oslash (R + \varepsilon)$ and $R = S \oslash (L + \varepsilon)$ in order to prevent the occurrence of a zero-denominator case, it is important to minimize the $\varepsilon$ value, thus avoiding a zero value.

### 2.2. Local Gradient Relative Deviation

In the statistical realm, the standard deviation function is used as a metric to gauge the coherence of a set of data. The image's local variation signifies the gradient characteristic of the image, while its deviation indicates the interconnection between neighboring pixels' changes in the immediate vicinity. Hence, the local deviation of the image affords a basis for discerning between texture (indicating a weak correlation) and structure (indicating a strong correlation).

Assume that the deviation of local change captured from image $S$ is $V_{x/y}$. The expression for $V_{x/y}$ can be represented using the subsequent formula:

$$V_{x/y} = \nabla_{x/y} G - \frac{1}{|\Omega|} \sum_{\Omega} \nabla_{x/y} G \tag{3}$$

where $\nabla_{x/y} G$ denotes the gradient in both the horizontal and vertical directions. $\Omega$ represents the size of the local image size $r \times r$ (generally, $r \times r$ is $3 \times 3$). To highlight the difference between structure and texture, we transform the formula into relative deviation:

$$V'_{x/y} = \frac{\nabla_{x/y} G}{\frac{1}{|\Omega|} \sum_{\Omega} \nabla_{x/y} G + \varepsilon} \tag{4}$$

In order to avoid a denominator of 0, the value of $\varepsilon$ should be as low as possible. The relative deviation of image local gradients can explain structural and texture smoothing properties. The instructions are as follows:

We assume that the average local gradient is $\overline{\nabla G} = \frac{1}{|\Omega|} \sum_{\Omega} \nabla G$.

Case 1. Assuming that the image is locally smooth, then gradient $\nabla G \approx 0$ and $\overline{\nabla G} \approx 0$ are the relative deviation of local variation $V' \approx 0$.

Case 2. Suppose that drastic changes occur locally in the image and that gradient, $\nabla G$ changes faster than $\overline{\nabla G}$; then, $\overline{\nabla G} > 0$ and the relative deviation of local changes is $V' \gg 1$.

Case 3. Suppose that the local change of the image is slow and the value of $\nabla G > 0$ is small; then, $\nabla G \approx \overline{\nabla G}$, $V' \approx 1$.

The local variation of an image represents gradient features, the local variation deviation of an image reflects the relationship between image texture and structure, and the local relative deviation of an image highlights the features of image structure and texture. Therefore, the local gradient relative deviation is used as a constraint term in this paper.

## 3. Proposed Methods

This chapter clarifies the step size framework of the proposed method illustrated in Figure 1 and illustrates the working principle of the proposed method (See Sections 3.2 and 3.3, and Algorithm 1).

### 3.1. Space Transformation

The human visual system exhibits a higher sensitivity to changes in brightness compared to alterations in color. Therefore, it is important for us to process low-luminosity

images using the photometric channel. In addition, if each channel of an RGB image is corrected, it is difficult to guarantee that each channel is given an appropriate ratio while being raised or lowered, which leads to color distortion in the enhanced image. In this paper, the transformation from the RGB image space to the HSV image space is carried out due to the independent nature of the three channels in HSV color images. The HSV color image consists of three channels: hue (H), saturation (S), and brightness (V). In this paper, local brightness channels are extracted from HSV images and corrected in the next step.

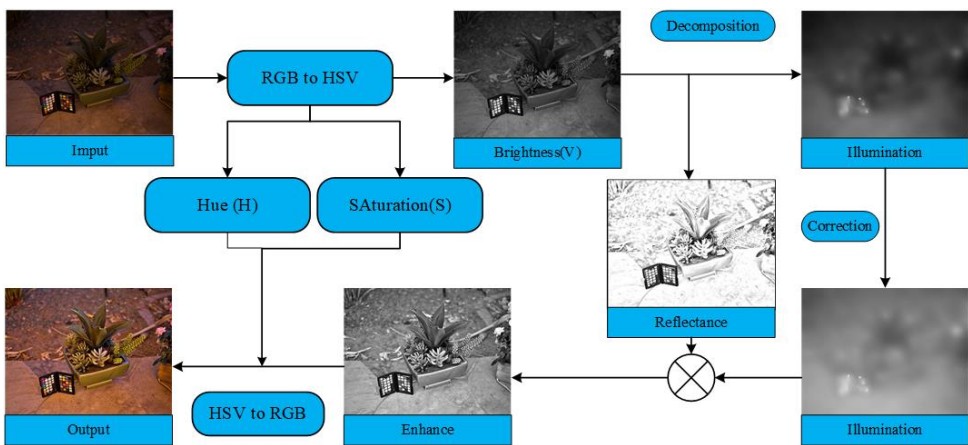

**Figure 1.** Illustration of proposed model.

### 3.2. The Proposed Model

In practice, noise in low-illumination images is unavoidable, so the Retinex model for the obtained actual images should be formulated as follows:

$$S = R \circ L + N \tag{5}$$

The local relative deviation of image highlights the characteristics of image structure and texture. Therefore, the relative deviation of the local gradients in the horizontal and vertical directions of the reflectance and illumination components of the object is used as a constraint term in this paper:

$$E_R(R) = \left\| \frac{\nabla R_x}{\left( \frac{1}{|\Omega|} \sum_{\Omega} \nabla R_x \right)^{\gamma_2}} \right\|_2^2 + \left\| \frac{\nabla R_y}{\left( \frac{1}{|\Omega|} \sum_{\Omega} \nabla R_y \right)^{\gamma_2}} \right\|_2^2 \tag{6}$$

$$E_L(L) = \left\| \frac{\nabla L_x}{\left( \frac{1}{|\Omega|} \sum_{\Omega} \nabla L_x \right)^{\gamma_1}} \right\|_P^P + \left\| \frac{\nabla L_y}{\left( \frac{1}{|\Omega|} \sum_{\Omega} \nabla L_y \right)^{\gamma_1}} \right\|_P^P \tag{7}$$

The $L_P$ norm $(0 < P < 1)$ has been demonstrated to produce a more sparse solution compared to the $L_1$ and $L_2$ norms, primarily because the $L_P$ norm approaches the behavior of the $L_0$ norm. Numerous applications in various related fields have confirmed its practicality. We constrain the illuminance and reflectance components by using a variational model and using the $L_2 - L_P$ norm. The formulation of the proposed model can be described as follows:

$$E(R, L) = \|R \circ L + N - S\|_2^2 + \alpha E_L(L) + \beta E_R(R) + \delta \|N\|_2^2 \tag{8}$$

where $\alpha$, $\beta$, and $\delta$ are regularized parameters.

The introduced function can be resolved through the iterative update of each variable, while the remaining variables estimated in the previous iteration can also be taken as constants. In this section, the article use alternating direction multiplier method (ADMM)

to decompose the objective function into three sub-problems and give the solution of the KTH iteration.

Sub-problems of $L$: Leaving out terms that are not connected to $L$, we optimize the problem as follows:

$$L_{k+1} = \arg\min_{L} \|R_k \circ L + N_k - S\|_2^2 + \alpha E_L(L) \tag{9}$$

To transform the problem into a conventional least squares problem, Equation (9) is restated as

$$l_{k+1} = \arg\min_{L} \|r_k \circ l + n_k - s\|_2^2 + \alpha E_L(l) \tag{10}$$

We transform Equation (10) as follows:

$$l_{k+1} = \arg\min_{L} \|D_{r_k} \cdot l + n_k - s\|_2^2 + \alpha \left( u_x \|\nabla L_x\|_2^2 + u_y \|\nabla L_y\|_2^2 \right) \tag{11}$$

where $r$ is the vectorization of the matrix $R$, and $D_{r_k}$ represents a diagonal matrix with $r$ as its term. The same notation is employed for the other matrices ($l$ and $n$ correspond to $L$ and $N$, respectively). But the optimization problem involving the $L_P$ norm is nonconvex and can pose challenges when solved directly without proper treatment. Therefore, this paper adopts the iterative reweighted least square method (IRLS) [23] to process $L_P$, $\|x\|_P^P = w \|x\|_P^P$, where $w = |x|^{p-2}$:

$$\begin{cases} w_x = |\nabla L_x + \varepsilon|^{p-2} \\ w_y = |\nabla L_y + \varepsilon|^{p-2} \end{cases} \tag{12}$$

$$\begin{cases} u_x = \left( \left| \frac{1}{\Omega} \sum_\Omega \nabla L_x \right|^{\gamma_1} \cdot |\nabla L_x|^{p-2} + \varepsilon \right)^{-1} \\ u_y = \left( \left| \frac{1}{\Omega} \sum_\Omega \nabla L_y \right|^{\gamma_1} \cdot |\nabla L_y|^{p-2} + \varepsilon \right)^{-1} \end{cases} \tag{13}$$

By taking the derivative of Equation (11) with respect to L and equating it to zero, we derive the following equation:

$$l_{k+1} = \left( D_{r_k}{}^T D_{r_k} + \alpha \left( G_x^T D_{u_x} G_x + G_y^T D_{u_y} G_y \right) \right) D_{r_k}{}^T (s - n_k) \tag{14}$$

where $G_x$ and $G_y$ are denoted as the Toplitz matrix of the discrete gradient operator with horizontal and vertical forward differences, and $D_{u_x}, D_{u_y}$ are diagonal matrices containing the weights $u_x$ and $u_y$, respectively.

Sub-problems for $R$: Ignoring terms that are not related to $R$, we optimize the problem as follows:

$$R_{k+1} = \arg\min_{R} \|R \circ L_{k+1} + N_k - S\|_2^2 + \beta E_R(R) \tag{15}$$

Similar to the previous formulation, $R$ can be solved as follows:

$$r_{k+1} = \arg\min_{R} \left\| D_{l_{k+1}} \cdot r + n_k - s \right\|_2^2 + \beta \left( v_x \|\nabla R_x\|_2^2 + v_y \|\nabla R_y\|_2^2 \right) \tag{16}$$

The solution is

$$r_{k+1} = \left( D_{l_{k+1}}{}^T D_{l_{k+1}} + \beta \left( G_x^T D_{v_x} G_x + G_y^T D_{v_y} G_y \right) \right) D_{l_{k+1}}{}^T (s - n_k) \tag{17}$$

Among them, $v_x = \left( \left| \frac{1}{\Omega} \sum_\Omega \nabla R_x \right|^{\gamma_2} + \varepsilon \right)^{-1}, v_y = \left( \left| \frac{1}{\Omega} \sum_\Omega \nabla R_y \right|^{\gamma_2} + \varepsilon \right)^{-1} D_{v_x}, D_{v_y}$ are diagonal matrices containing the weights $v_x$ and $v_y$, respectively.

Sub-problems for $N$: In order to optimize the problem while disregarding terms unrelated to $N$, we can proceed as follows:

$$N_{k+1} = \arg\min_R \|R_{k+1} \circ L_{k+1} + N - S\|_2^2 + \delta \|N\|_2^2 \tag{18}$$

In this paper, by differentiating the expression (18) and setting the derivative as 0, the solution of this problem is

$$N_{k+1} = (S - R_{k+1} \circ L_{k+1})/(1 + \delta) \tag{19}$$

where / represents element division.

$L$ and $R$ are updated iteratively until the cutoff condition $\|L_k - L_{k-1}\|/\|L_{k-1}\| \leq \varepsilon$ or $\|R_k - R_{k-1}\|/\|R_{k-1}\| \leq \varepsilon$ is satisfied. In order to solve the linear equation in this paper and make it converge quickly, a fast solver with preconditioned conjugate gradient (PCG) [24] is used to accelerate its speed. By setting the preconditions, the algorithm iterates through the preconditions, and the efficiency of the algorithm is improved by adjusting the order of the computations to obtain the exact solution more quickly. In the solution procedure, the solution of the last iteration is written in the form of a decomposition by negative gradients and conjugation conditions, so that the solution of the next iteration can be found quickly.

### 3.3. Illumination Correction

The last part of the algorithm in this chapter is the illumination component correction. To enhance the visibility of the input image, in this current undertaking, the illumination components are adjusted using gamma correction. The generation of the improved V-channel images was accomplished through

$$S' = R \circ L' \tag{20}$$

$$L' = L^{\frac{1}{\gamma}} \tag{21}$$

Following other scholars [8,9,11,13,14,16], $\gamma$ was empirically set to 2.2. Finally, we converted the enhanced HSV image into the RGB color space to obtain the final enhanced result, denoted as $S'$.

---

**Algorithm 1** Low-illumination image enhancement using local gradient relative deviation for Retinex model

---

**Input:** image $S$, parameter $\alpha, \beta, \gamma, \gamma_1, \gamma_2, \delta, \varepsilon$
     Maximum number of iterations $K$, cut-off condition $\varepsilon$
**Output:** illuminance component $L$ and reflectivity component $R$

1: Initialize $L_0 \leftarrow S$
2: **for** $k = 1$ **do**
3:     calculate weights $u_{x/y}$ in Equation (12)
4:     to modify the update of $L_k$ using Equation (14)
5:     **if** $k = 1$ **then**
6:        $R_0 = S/L_1$
7:     **end if**
8:     calculate weights $v_{x/y}$ in Equation (17)
9:     to modify the update of $R_k$ using Equation (17)
10:    update $N_k$ using Equation (19)
11:    **if** $(L_k - L_{k-1})/L_{k-1} \leq \varepsilon$ or
      $(R_k - R_{k-1})/R_{k-1} \leq \varepsilon$ **then**
12:      break
13:    **end if**
14: **end for**
15: **return** $L' = L^{\frac{1}{\gamma}}$ $S' = R \circ L'$

---

## 4. Experimental Results

In this section, the performance of the proposed method is evaluated. First, we present the experimental empirical parameters of the ensemble. Second, we compare the performance of the proposed method with state-of-the-art low-illumination image enhancement techniques and analyze the results obtained from the comparison. To fully evaluate the methods presented in this paper, all experiments were performed on MATLAB R2019a and the program was run on a Windows 7 server with 8 GB of memory and a 3.5 GHz CPU. Meanwhile, to fully evaluate the proposed method, this paper takes images of different scenarios for testing. Image data include LOL dataset [21], AID dataset [25], VV dataset [26], SCIE dataset [27], and TGRS-HRRSD dataset [28].

In this trial, after both qualitative and quantitative analyses, the values of the experimental encounter factors have been established as follows: $\alpha$ at 0.001, $\beta$ at 0.0001, $\gamma$ at 2.2, $\gamma_1$ at 1.5, $\gamma_2$ at 0.05, $\delta$ at 0.001, $\varepsilon$ at 0.0001. When the magnitude of $\alpha$ becomes excessively high, the separated component $L$ lacks information or experiences partial loss of contour details. Conversely, when $\alpha$ takes on excessively low values, component $L$ exhibits inadequate smoothness. At considerably elevated values of $\beta$, the resolved component $R$ appears to be over-smoothed, causing the finer details of $R$ to become blurred. In turn, when an excessively small value of $\beta$ is assumed, the decomposition results are similar to the case where $\beta$ is set to 0.0001.

When $\gamma_1$ assumes excessively small values, the smoothness of the separated component $L$ diminishes; conversely, when $\gamma_1$ becomes overly large, contour information within the decomposed $L$ component becomes absent. A significantly elevated value of $\gamma_2$ results in an excessively smooth decomposed component $R$, leading to the loss of detailed information. When $\gamma_2$ takes on excessively small values, the effect of decomposed component $R$ remains similar to that when $\gamma_2$ is set to 0.05. The solution for $N$ is linked to the values of $R$, $L$, and $\delta$. Larger $\delta$ values introduce noticeable noise into the decomposed $R$. Conversely, smaller $\delta$ values yield $R$ outcomes with reduced noise, although some noise remains due to the relatively greater influence of $R$ and $L$ within the $N$ solution.

The presence of $\varepsilon$ in the denominator serves to prevent division by zero. Consequently, minimizing $\varepsilon$ is preferable. Excessively large $\varepsilon$ values impact the denominator and directly alter the decomposition results. Conversely, when $\varepsilon$ becomes exceedingly small, the outcome is akin to using $\varepsilon$ values of 0.0001, and the solution process converges.

In this article, the proposed method was compared with the Joint Internal and External Prior (JieP) model based on Retinex [14], the Retinex enhancement method (SEM) based on robustness [15], the improved model based on semi-decoupling-decomposition (SDD) [13], the joint enhancement and denoising model based on sequence decomposition (JED) [16], and the low-light enhancement plug-and-play Retinex Model (LUPP) [18], and other methods.

### 4.1. Decomposition Evaluation

In Figure 2, the reflectance of objects resolved by Method 5 [18] is generally dark and many details are lost. According to the decomposition results of Method 1 [16] and Method 2 [15], a part of the reflectance information of the object is lost and the color is significantly excessive. In contrast to methods 3 [14] and 4 [13], the proposed method can ensure maximum color information while highlighting reflectivity details. By comprehensive comparison, the method in this paper has advantages.

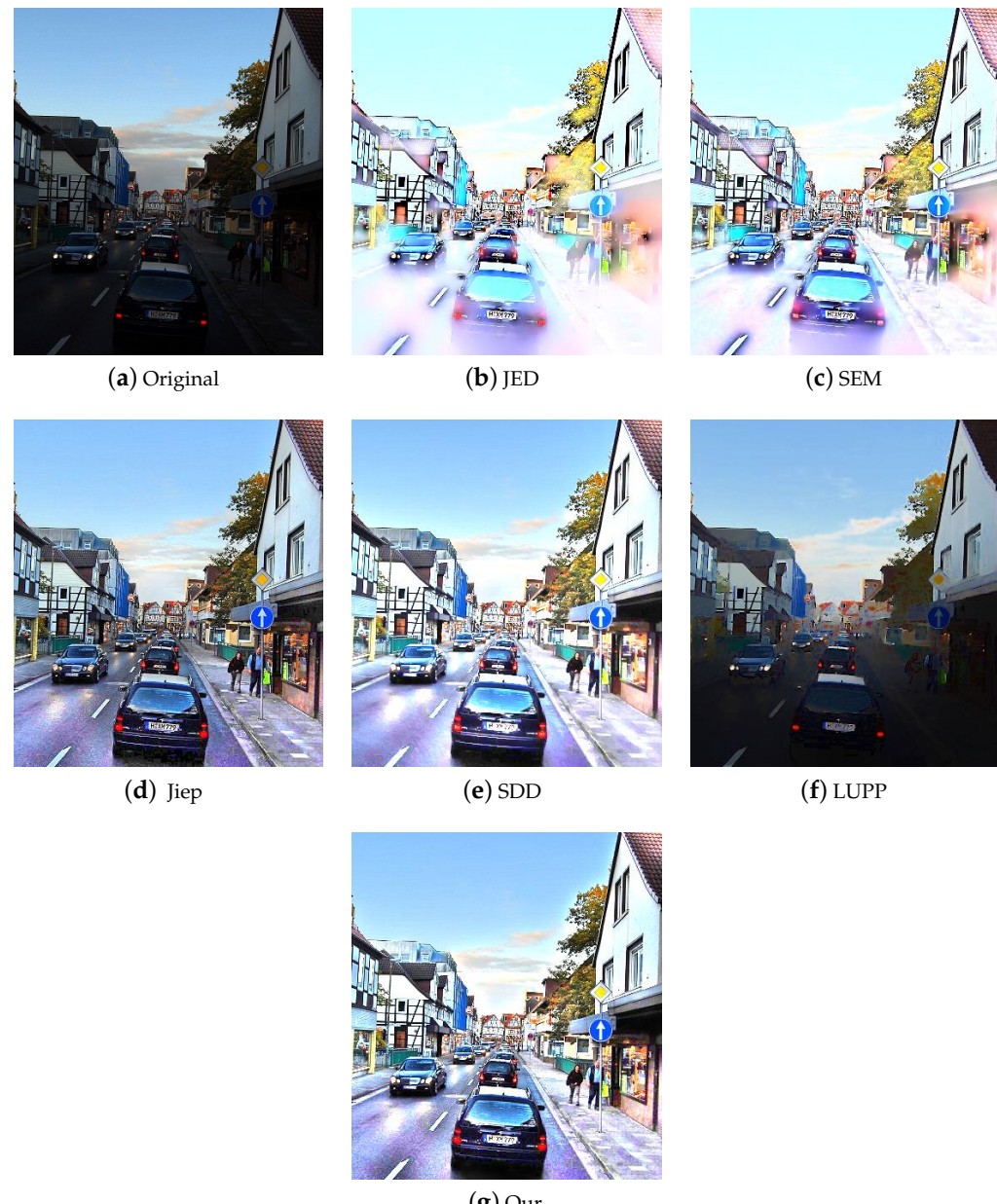

**Figure 2.** Object reflectance, from left to right: (**a**) Original; (**b**) JED; (**c**) SEM; (**d**) Jiep; (**e**) SDD; (**f**) LUPP; (**g**) Our model.

In Figure 3, the illumination component decomposed by the proposed method is significantly smoother, while the illumination components decomposed by Methods 1 [16], 2 [15], and 3 [14] are weaker than the other Methods. The intensity of the illuminant component of the decomposition by Method 5 [18] is low, and the error of the illuminant component is large. Compared to Method 4 [13], the proposed method has better smoothness. By comprehensive comparison, the method in this paper shows clear advantages.

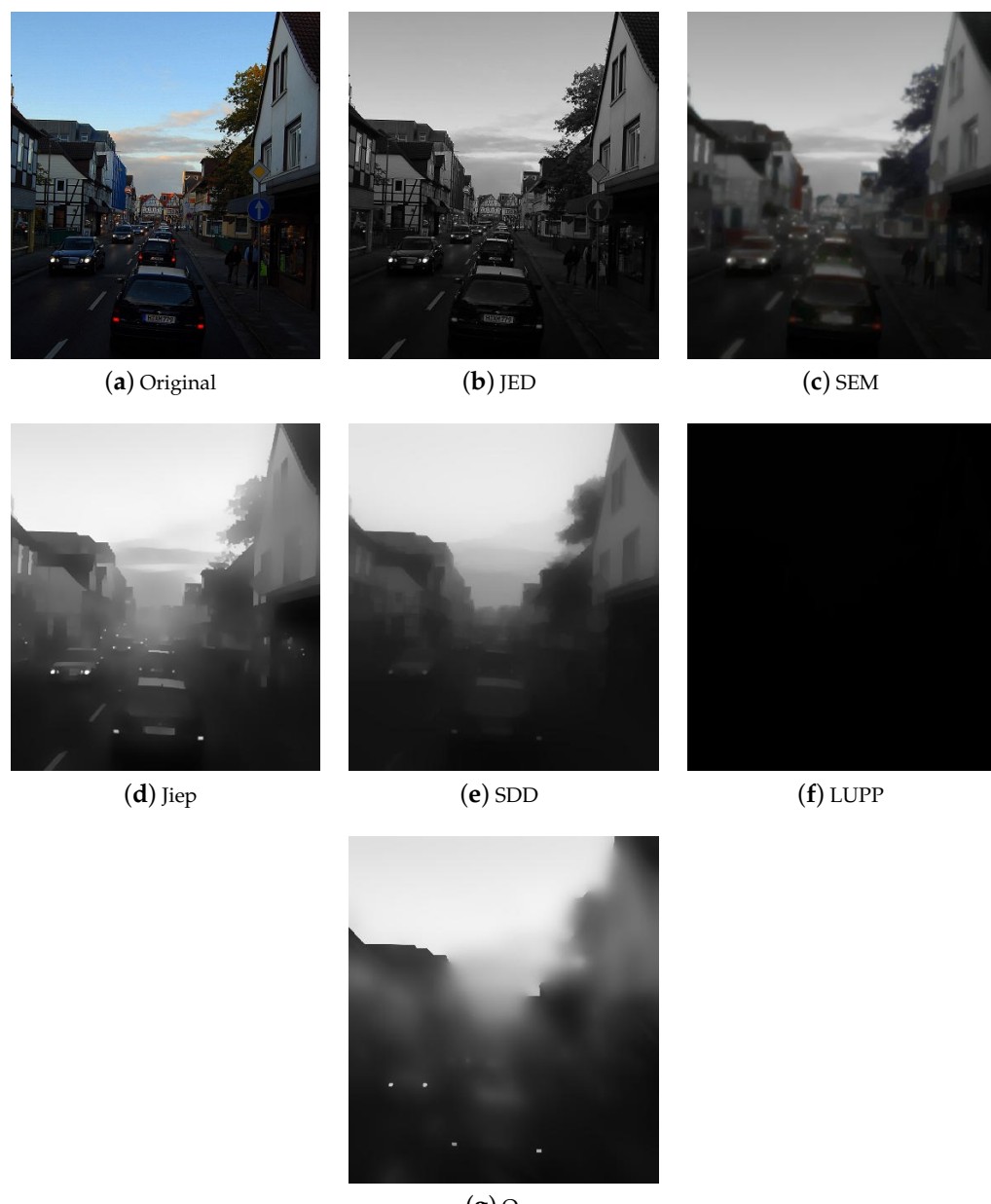

**Figure 3.** Object illuminance component from left to right: (**a**) Original; (**b**) JED; (**c**) SEM; (**d**) Jiep; (**e**) SDD; (**f**) LUPP; (**g**) Our model.

### 4.2. Objective Evaluation

To conduct an objective evaluation of the quality of the enhanced results, this paper employed five widely recognized image quality assessment (IQA) metrics. Tables 1–5 present the average values of the IQA metrics for the four datasets(The bold data in tables represents the best value for its corresponding metric in that data set). The five IQAs include unreferenced/blind assessment methods and fully referenced assessment methods.

In the natural image quality evaluator (NIQE) [29] approach, some image patches are selected as training data based on local features, and the model parameters are obtained by fitting a generalized Gaussian model to the features, which are described by a multivariate Gaussian model. During the evaluation process, the distance between the parameters of the image feature model to be evaluated and the parameters of the pre-established model are used to determine the image quality. AutoRegressive-based image sharpness metric (ARISM) [30] works by separately measuring the energy and contrast differences of the AR model coefficients at each pixel and then computing the image sharpness with



percentile pooling to infer the overall quality score. Colorfulness-based PCQI (CPCQI) [31] evaluates the perceptual distortion between the enhanced image and the input image via three aspects: mean intensity, signal strength, and signal structure. Visual information fidelity (VIF) [32] combines the natural image statistical model, image distortion model, and human visual system model to calculate the mutual information between the image to be evaluated and the reference image, to measure the quality of the image to be evaluated. Lightness order error (LOE) [10] evaluates the image quality by separately extracting the brightness of the original image and the enhanced image, and then calculating the relative magnitude difference in brightness from each pixel. The higher the values of CPCQI and VIF, the better the image quality. Lower values of NIQE, LOE, and ARISM indicate better image quality.

**Table 1.** Results of quantitative comparison on LOL data set (90 charts).

| Methods | JED | SEM | Jiep | SDD | LUPP | Our |
|---------|-----|-----|------|-----|------|-----|
| NIQE  | 6.22   | 5.82   | 6.34     | 5.50   | 7.10   | **5.20**   |
| CPCQI | 0.966  | 0.971  | 0.998    | 0.975  | 0.998  | **0.999**  |
| VIF   | 7.427  | 7.788  | 12.109   | 12.686 | 15.919 | **17.068** |
| ARISM | **0.740** | 0.786 | 0.869 | 0.978  | 1.055  | 0.974 |
| LOE   | 354.32 | 396.51 | **312.52** | 440.16 | 576.37 | 430.3 |

**Table 2.** Quantitative comparison results on SCIE data sets (49 charts).

| Methods | JED | SEM | Jiep | SDD | LUPP | Our |
|---------|-----|-----|------|-----|------|-----|
| NIQE  | 4.35   | 4.35   | 3.84     | 3.92   | 3.76   | **3.73**  |
| CPCQI | 0.989  | 0.991  | **1.074** | 0.994 | 0.996  | 0.997 |
| VIF   | 2.090  | 2.123  | 2.581    | 3.636  | **4.136** | 3.907 |
| ARISM | 1.190  | 1.198  | 2.549    | 1.178  | 1.189  | **1.177** |
| LOE   | 904.42 | 905.42 | **891.48** | 915.4 | 917.93 | 901.35 |

**Table 3.** Quantitative comparison results on VV data set (24 charts).

| Methods | JED | SEM | Jiep | SDD | LUPP | Our |
|---------|-----|-----|------|-----|------|-----|
| NIQE  | 3.61   | 3.51   | 3.00     | 3.16   | **2.93** | 3.03 |
| CPCQI | 0.985  | 0.987  | 0.988    | 0.988  | 0.99   | **0.991** |
| VIF   | 1.185  | 1.216  | 1.43     | 1.85   | 1.872  | **1.892** |
| ARISM | 1.231  | 1.238  | 2.577    | 1.230  | 1.241  | **1.156** |
| LOE   | 527.11 | 541.69 | **416.42** | 504.57 | 564.94 | 497.45 |

**Table 4.** Quantitative comparison results on AID data set (90 charts).

| Methods | JED | SEM | Jiep | SDD | LUPP | Our |
|---------|-----|-----|------|-----|------|-----|
| NIQE  | 4.93   | 4.28   | 3.55     | 3.93   | 3.45   | **3.42** |
| CPCQI | 0.996  | 0.997  | **1.079** | 0.996 | 0.998  | 0.997 |
| VIF   | 2.261  | 2.488  | 4.049    | 5.092  | **5.764** | 5.181 |
| ARISM | 1.300  | 1.293  | 1.291    | 1.285  | 1.301  | **1.280** |
| LOE   | 375.64 | 321.32 | 397.91   | 339.83 | 270.32 | **268.61** |

**Table 5.** Quantitative comparison results on TGRS-HRRSD data sets (140 charts).

| Methods | JED | SEM | Jiep | SDD | LUPP | Our |
|---------|-----|-----|------|-----|------|-----|
| NIQE  | 4.18   | 3.98   | 3.67     | 3.64   | 3.81   | **3.53** |
| CPCQI | 0.991  | 0.993  | **1.158** | 0.994 | 0.996  | 0.995 |
| VIF   | 3.514  | 3.880  | 5.951    | 7.343  | 8.427  | **9.284** |
| ARISM | 1.227  | 1.236  | 2.612    | 1.253  | 1.271  | **1.174** |
| LOE   | 698.86 | 681.32 | 747.16   | 719.33 | **598.96** | 654.27 |

In the data in Table 1, the proposed method demonstrates superior performance compared to other algorithms in terms of CPCQI, NIQE, and VIF. On the other hand, both ARISM and LOE exhibit better results than Method 5 [18]. In the data in Table 2, the proposed algorithm's NIQE and ARISM outperform the other methods, while CPCQI and VIF are only second to Methods 3 [14] and 5 [18], respectively, and the LOE method is also second to Method 3 [14]. In the data in Table 3, the CPCQI, VIF, and ARISM metrics of the introduced algorithm are slightly better than the other methods; In terms of NIQE, the algorithm is better than Methods 1 [16], 2 [15], and 4 [13], close to Method 3 [14], and slightly weaker than Method 5 [18]. It is second only to Method 3 [14] in terms of LOE. In the data in Table 4, NIQE, LOE and ARISM demonstrate superior performance compared to the remaining methods, whereas CPCQI exhibits slightly lower performance than Method 5 [18]. In Table 5, the proposed method outperforms the other algorithms in NIQE, VIF, and ARISM, with LOE second only to Method 5 [18], and CPCQI slightly weaker than Method 5 [18]. After comparison on four datasets, the proposed method outperforms the others.

*4.3. Subjective Evaluation*

Figures 4–11 contain the enhanced results of different images using different methods.

In the improved outcomes presented in Figure 4, the central lines of the illustrations and the books, as enhanced through techniques 1 [16] and 2 [15], exhibit excessive brightness (contained within the left-hand rectangular enclosure). Nonetheless, these enhancements lack a sense of naturalness and display an overly pronounced smoothness, leading to a loss of sharp details in both the lines and the intricate features of the doll's nose (as depicted within the right-hand rectangular enclosure). The overall tone of the resulting image from the application of Method 3 [14] appears significantly darker, whereas the enhancement achieved by the method introduced in this paper renders the cartoon doll significantly superior in terms of overall textural quality, surpassing the results of Method 3 [14], Method 4 [13], and Method 5 [18]. It is important to note that, in contrast to method 5 [18], the technique presented in this study exhibits a discernible noise reduction effect while still preserving intricate textural details (as is evident within the restricted region of the small red rectangle on the right).

Within the enhancement outcomes depicted in Figure 5, the evident smoothing effects from Methods 1 [16], 2 [15], and 4 [13] lead to a conspicuous blurring of intricate flower textures and chair details, observable within the two highlighted red rectangular sections within the illustration. In addition, the enhancement results of Method 2 [15] show a slight color distortion. In contrast, the augmentation of Method 3 [14] exhibits a relatively darker appearance among the various techniques. Contrasting with the approach introduced in this paper, the enhancement results of Method 5 [18] distinctly reveal noticeable noise within the chair segment (as indicated by the red rectangular region on the right). It is worth noting that the method presented in this study demonstrates a discernible noise reduction capability while preserving intricate details.

Displayed in Figure 6, the enhanced outcomes arising from Method 1 [16] and Method 2 [15], although yielding a heightened overall luminosity in comparison to alternative techniques, suffer from an excess of smoothing that detrimentally affects the intricate details within the scenery, leading to a loss of clarity as demonstrated within the enclosed red rectangular region on the right. The combined visual impact of Methods 4 [13] and 5 [18] closely approximates that of the algorithm introduced herein, yet the latter significantly outperforms the former two in low-light scenarios, as evidenced by the red rectangle on the left. In turn, the enhancement results stemming from Method 3 [14] are biased towards general darkness and exhibit weaker refinement compared to the other algorithms. Through a comprehensive evaluation, it becomes clear that the proposed method exhibits superior performance when measured against alternatives.

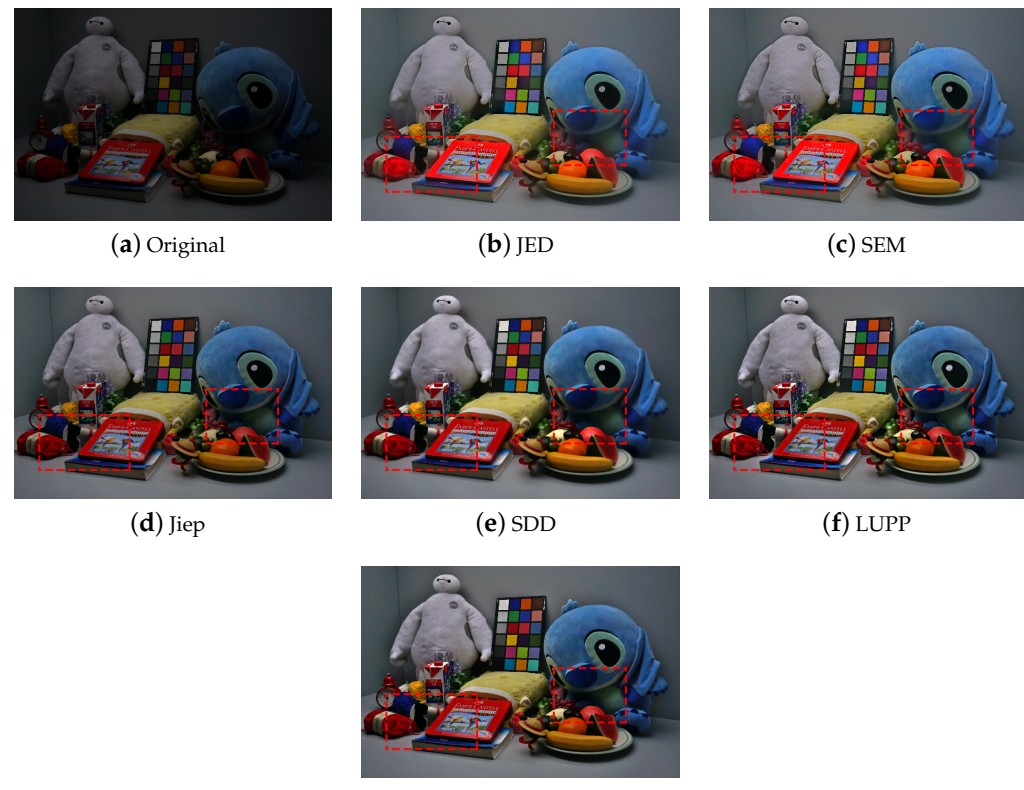

**Figure 4.** Cartoon character image enhancement results from left to right: (**a**) Original; (**b**) JED; (**c**) SEM; (**d**) Jiep; (**e**) SDD; (**f**) LUPP; (**g**) Our model.

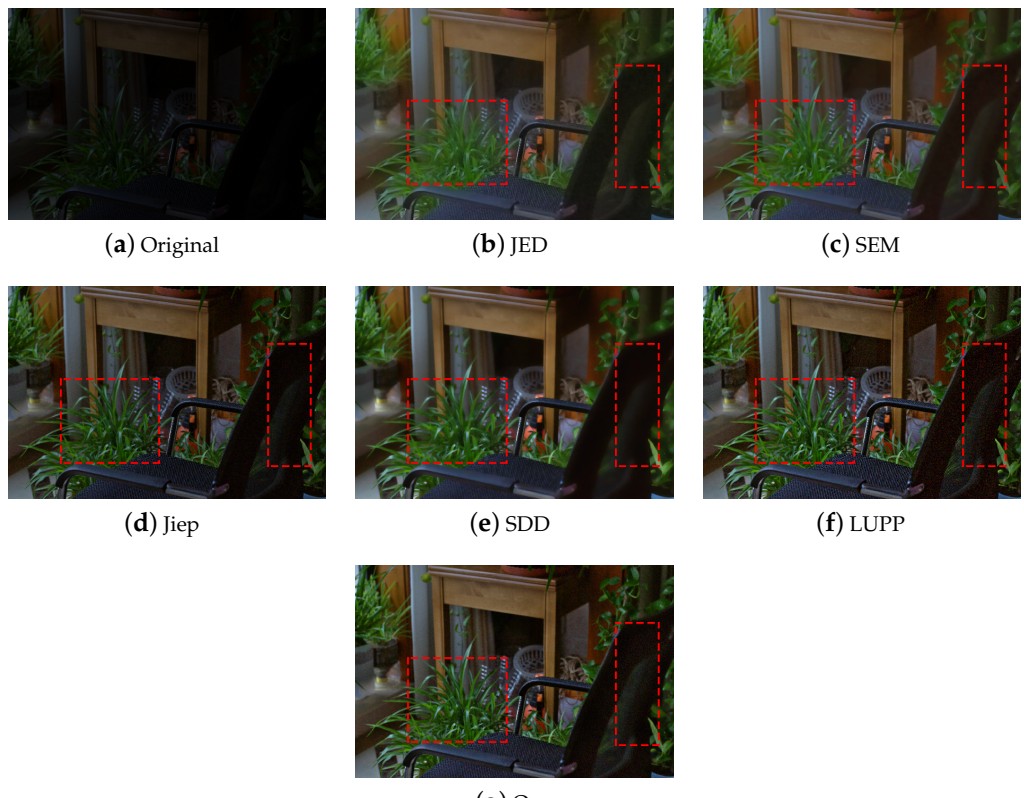

**Figure 5.** Indoor image enhancement results from left to right: (**a**) Original; (**b**) JED; (**c**) SEM; (**d**) Jiep; (**e**) SDD; (**f**) LUPP; (**g**) Our model.

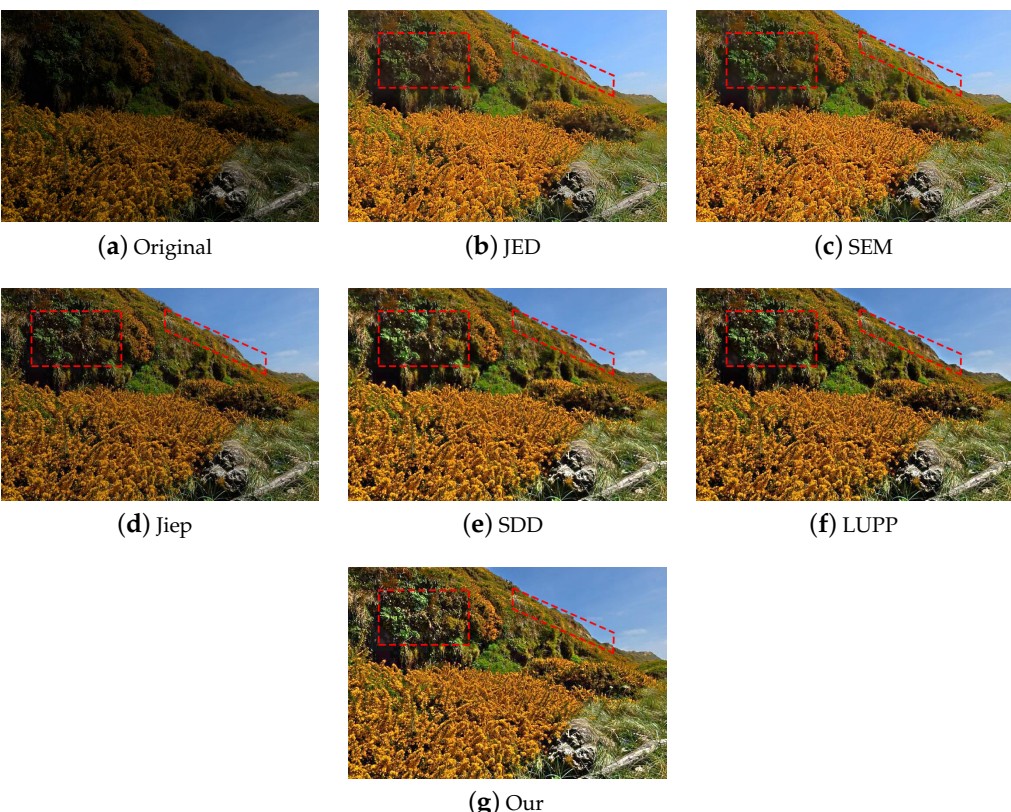

**Figure 6.** Landscape image enhancement results from left to right: (**a**) Original; (**b**) JED; (**c**) SEM; (**d**) Jiep; (**e**) SDD; (**f**) LUPP; (**g**) Our model.

As illustrated in Figure 7, the treatment of the shopping mall staircases and signage using Methods 1 [16] and 2 [15] yields an overly conspicuous smoothness, resulting in obscured stairway details and blurred representations of store signboards—both instances can be observed within the enclosed red rectangular compartment in figure. Simultaneously, the interiors of the stores in the enhancement outcomes of Methods 1 [16] and 2 [15] suffer from pronounced blurring, resulting in reduced sharpness when compared to the alternative techniques. In addition, the overall contrast of store details within Methods 1 [16] and 2 [15] falls short when compared to other approaches. The enhanced outcome of Method 3 [14] tends to exhibit an overall darker appearance. In the case of Method 4 [13], detail retention slightly lags behind that of Algorithm 5 [18] and the approximations delineated in this paper, which is particularly evident within the upper left stairway region enclosed by the red rectangle. Moreover, in terms of intricate color processing, the method presented in this paper triumphs over Methods 4 [13] and 5 [18]. This triumph is particularly evident in the color preservation of detailed elements, such as the signboard depicted within the figure. In the enhancement results of the other methods, the right segment of the signboard experiences color distortion, while the method introduced in this paper remarkably preserves the original image color.

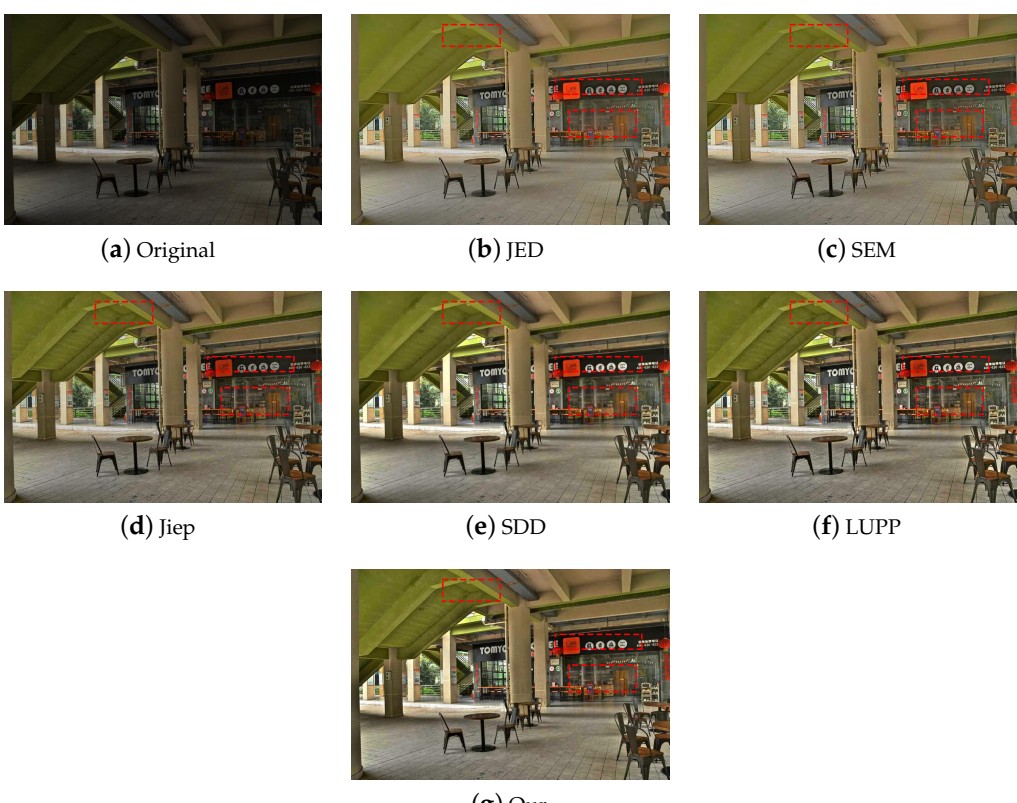

**Figure 7.** Mall image enhancement results from left to right: (**a**) Original; (**b**) JED; (**c**) SEM; (**d**) Jiep; (**e**) SDD; (**f**) LUPP; (**g**) Our model.

In the results shown in Figure 8, the overall enhancement in image brightness achieved by employing Methods 1 [16] and 2 [15] surpasses the performance of alternative approaches. Nonetheless, these methods fall short in terms of visual dynamic range when compared to other algorithms. Additionally, an excess of smoothing is evident in the images, leading to a lack of emphasis on intricate details within the components like trees, squares, and houses (marked by three red rectangular boxes). Method 3 [14], although it manages to maintain these fine details, results in an overall darker image. In contrast, when measured against Methods 4 [13] and 5 [18], the proposed method in this study strikes a balance by maintaining a certain level of smoothness while preserving crucial details. It is notable that the detail preservation achieved by Method 4 [13] is not as potent as that of Method 5 [18] or the technique introduced in this paper (highlighted in the upper left red rectangular box). Comparatively, Method 5 [18] displays evident noise in specific portions of the image (as seen in the middle red rectangular frame containing trees). Therefore, the method proposed in this paper demonstrates an ability to mitigate noise to a considerable extent. Moreover, when stacked against alternative algorithms, the proposed method exhibits superior detail retention while effectively suppressing noise. As a result, the proposed algorithm boasts an overall advantage in this context.

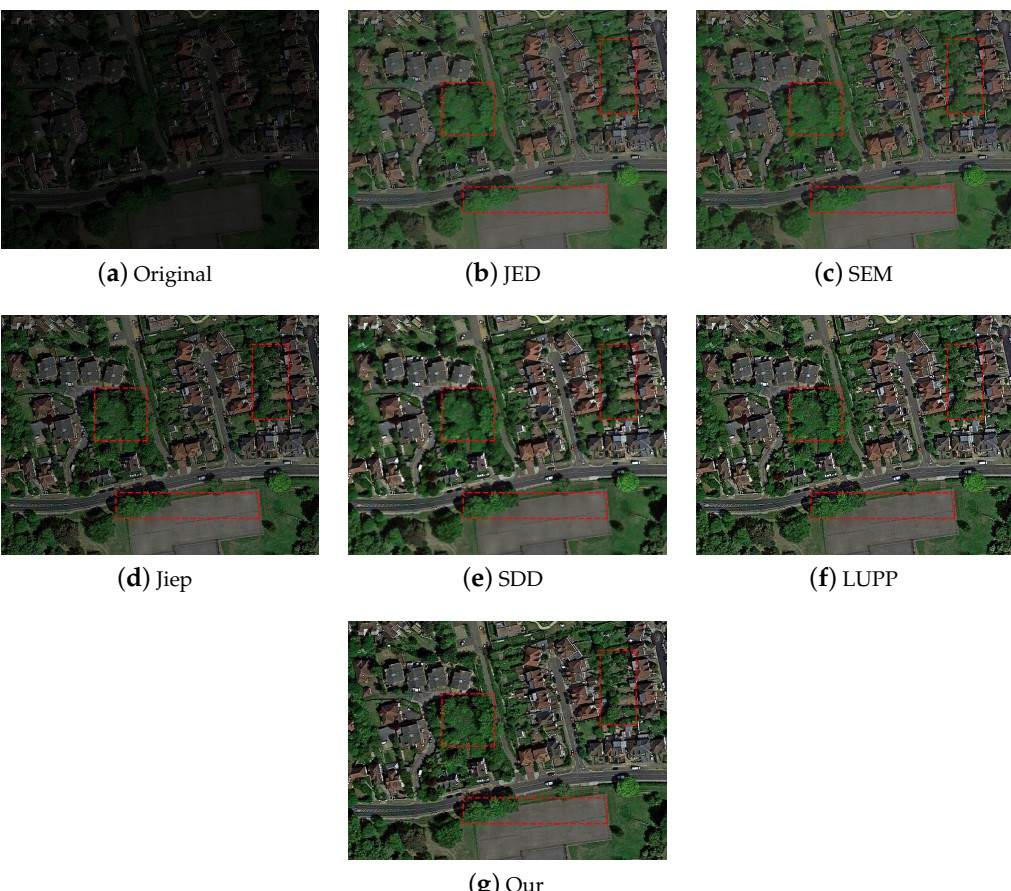

**Figure 8.** Remote sensing image 1 enhancement results from left to right: (**a**) Original; (**b**) JED; (**c**) SEM; (**d**) Jiep; (**e**) SDD; (**f**) LUPP; (**g**) Our model.

In Figures 9–11, the enhancement of image dynamic range achieved through Methods 1 [16] and 2 [15] falls behind that of alternative algorithms. Moreover, an excessive application of image smoothing leads to the unfortunate consequence of losing intricate details within the images. When juxtaposed against Methods 3 [14], 4 [13], and 5 [18], the method proposed in this study adeptly maintains a certain degree of smoothness while safeguarding essential details. Comparatively, Method 3 [14] results in darker enhanced images when contrasted with Methods 4 [13], 5 [18], and the techniques introduced within this paper. Method 5 [18], however, reveals conspicuous noise in specific regions of the images (highlighted by red rectangular boxes in the upper left of Figure 9, upper right of Figure 10, and other sections in Figure 11). In light of this, the approach outlined in this paper demonstrates a noteworthy capability in mitigating noise to a certain extent.

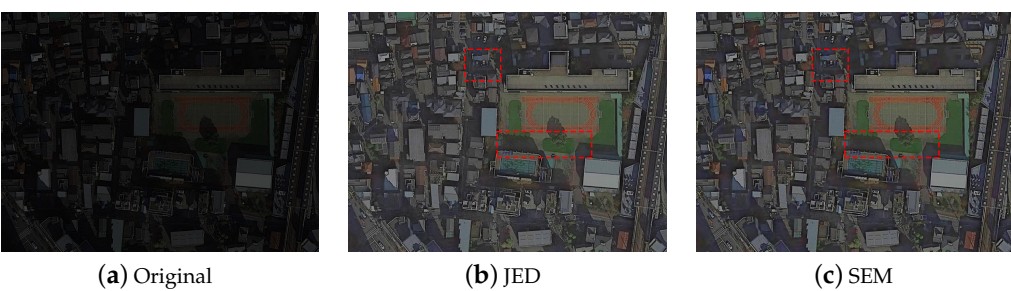

**Figure 9.** *Cont*.

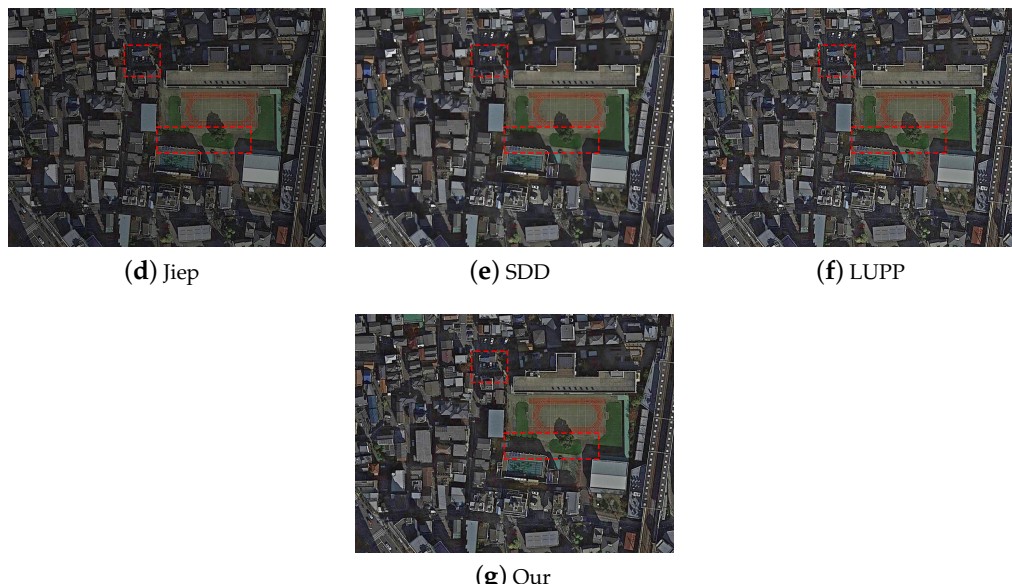

**Figure 9.** Remote sensing image 2 enhancement results from left to right: (**a**) Original; (**b**) JED; (**c**) SEM; (**d**) Jiep; (**e**) SDD; (**f**) LUPP; (**g**) Our model.

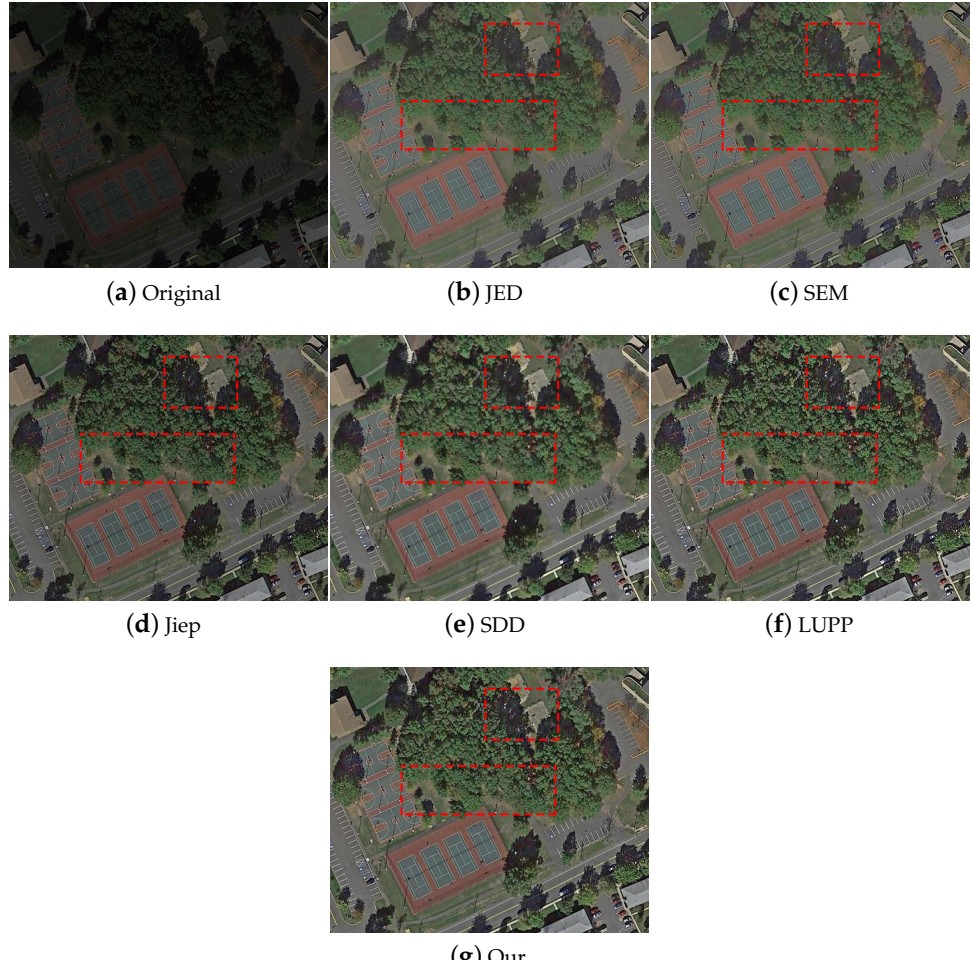

**Figure 10.** Remote sensing image 3 enhancement results from left to right: (**a**) Original; (**b**) JED; (**c**) SEM; (**d**) Jiep; (**e**) SDD; (**f**) LUPP; (**g**) Our model.

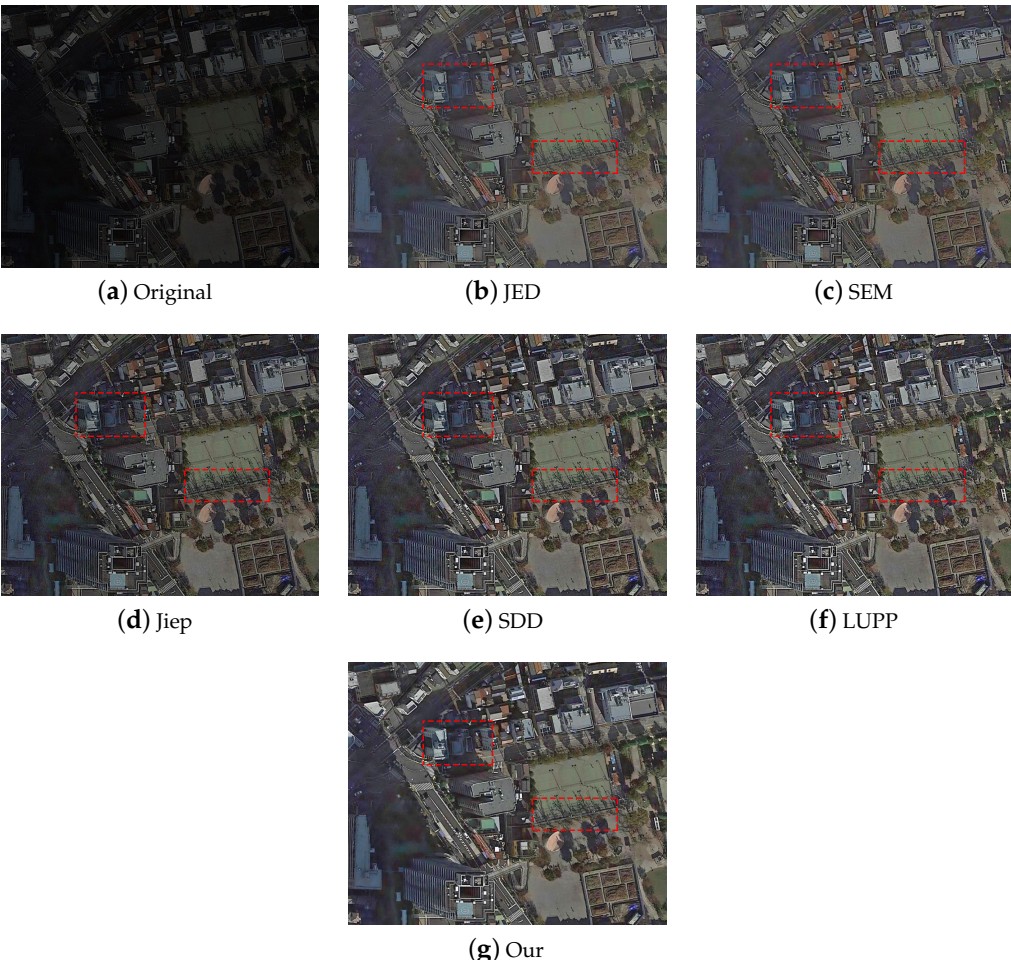

**Figure 11.** Remote sensing image 4 enhancement results from left to right: (**a**) Original; (**b**) JED; (**c**) SEM; (**d**) Jiep; (**e**) SDD; (**f**) LUPP; (**g**) Our model.

### 4.4. Real Remote Sensing Image Data

To assess the effectiveness of the proposed algorithm in practical scenarios, real images taken in the laboratory are used for validation and the average IQA values are listed in Table 6.

**Table 6.** Quantitative comparison results of real data sets (40 figs).

| Methods | JED | SEM | Jiep | SDD | LUPP | Our |
|---|---|---|---|---|---|---|
| NIQE | 7.56 | 6.38 | 5.63 | 6.15 | 6.35 | **5.32** |
| CPCQI | 0.992 | 0.997 | 0.984 | 0.989 | 0.993 | **0.999** |
| VIF | 2.638 | 3.452 | 6.573 | 5.959 | 5.542 | **7.835** |
| ARISM | **0.883** | 0.976 | 0.993 | 1.154 | 0.986 | 1.068 |
| LOE | 653.63 | **525.13** | 694.53 | 784.28 | 886.78 | 958.42 |

In Table 6, NIQE, VIF, and CPCQI of the proposed method outperform the other methods, while LOE and ARISM are slightly weaker.

To assess the visual outcomes achieved by implementing the proposed algorithm in real-world applications, this article adopt real images taken in the laboratory for validation and present the enhancement results of the different methods in Figures 12 and 13.

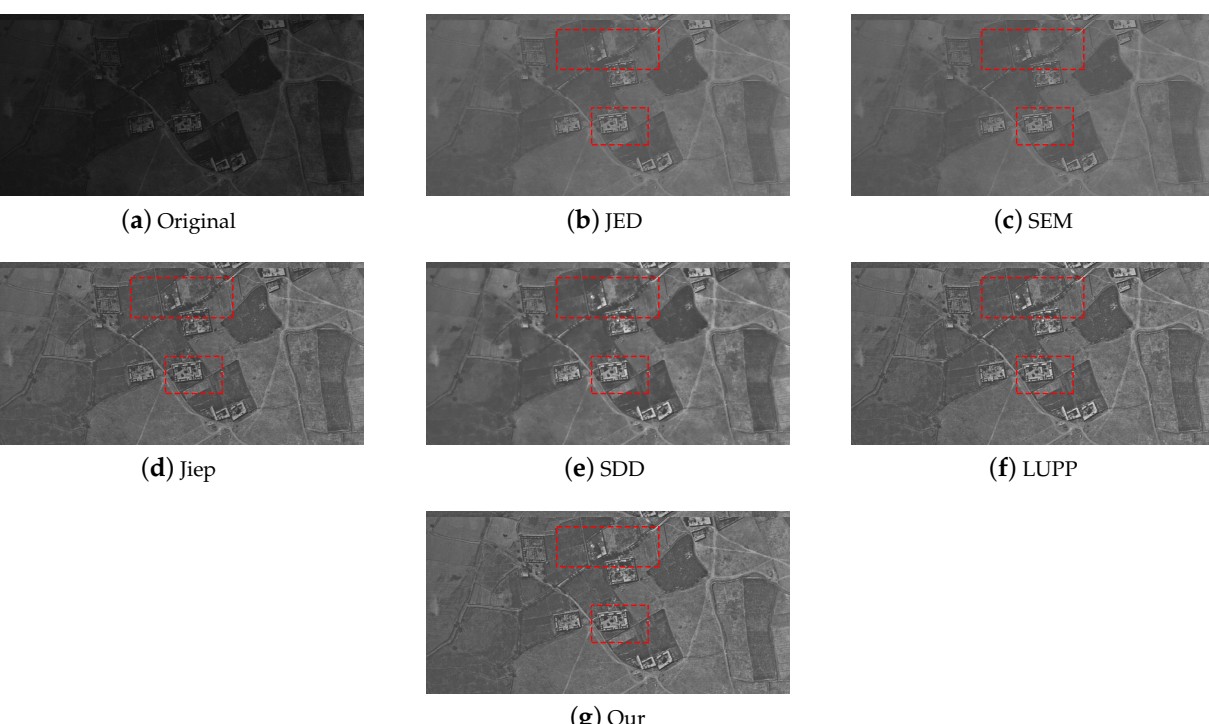

**Figure 12.** Real image data 1 enhancement results from left to right: (**a**) Original; (**b**) JED; (**c**) SEM; (**d**) Jiep; (**e**) SDD; (**f**) LUPP; (**g**) Our model.

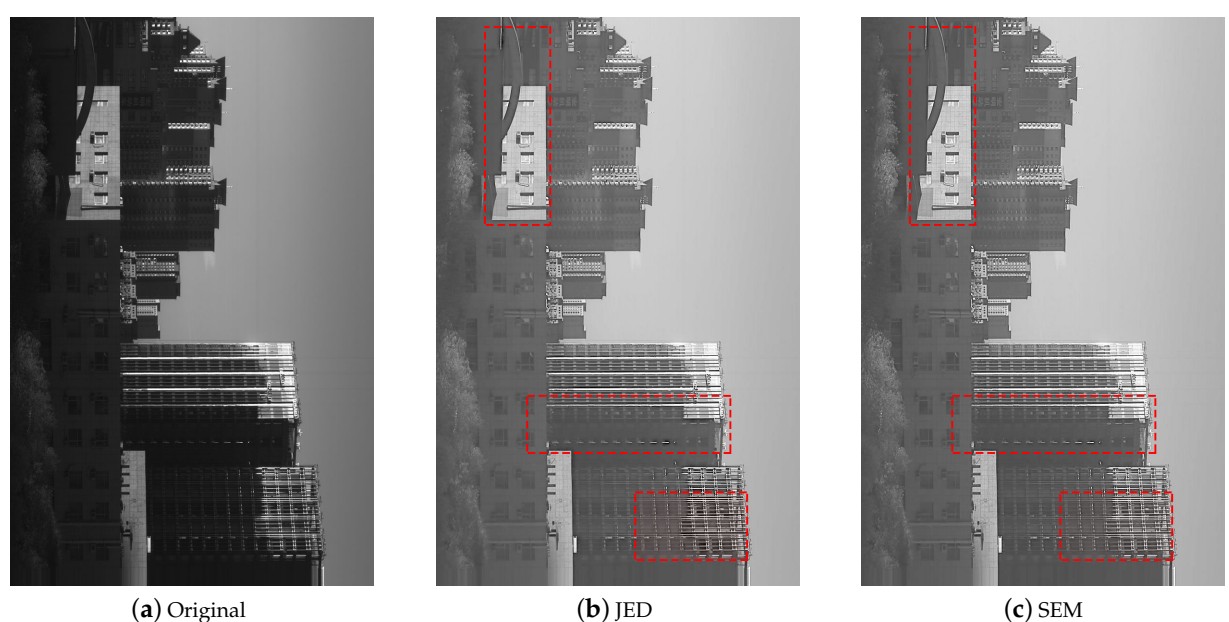

**Figure 13.** *Cont.*

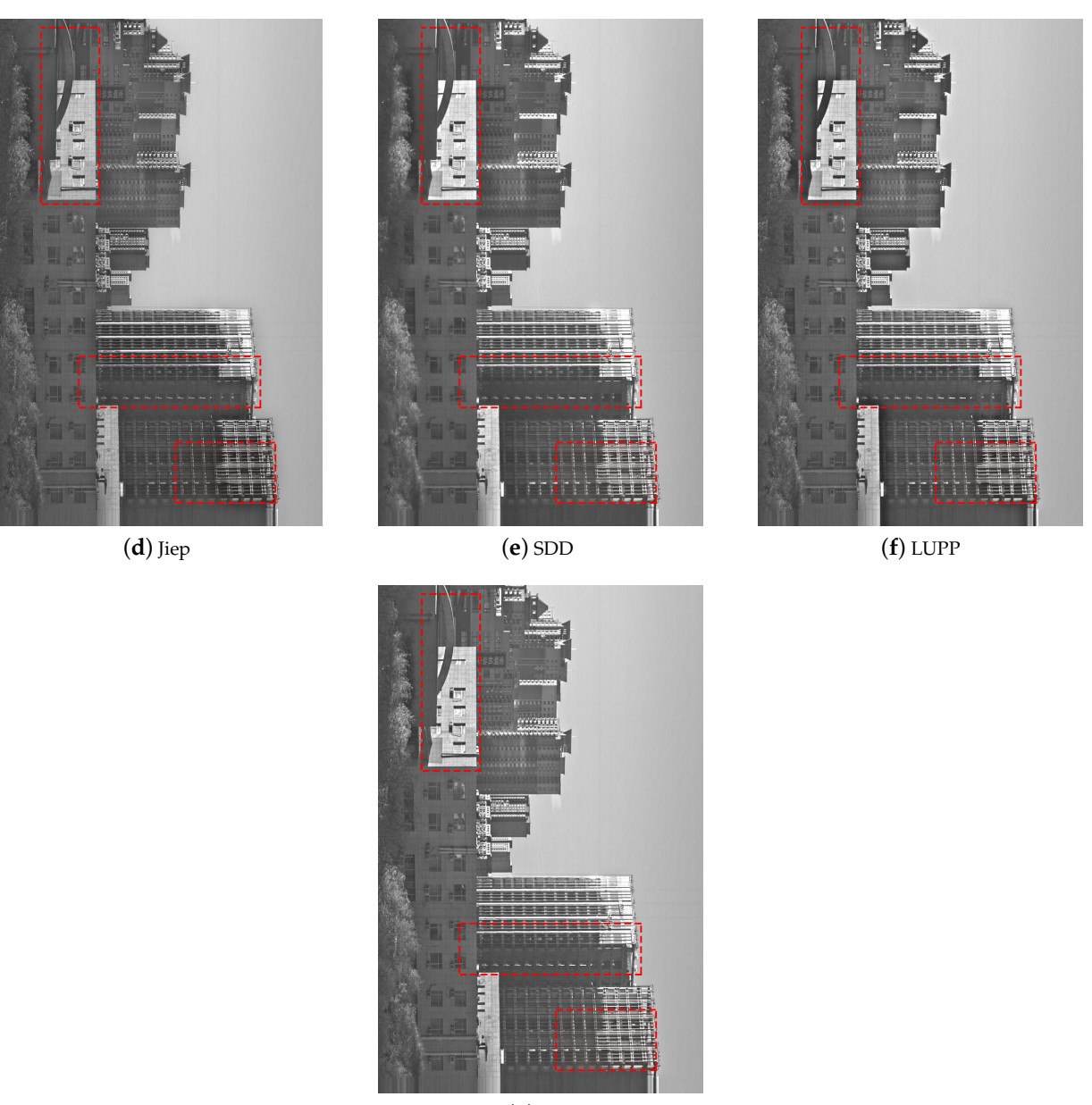

**Figure 13.** Real image data 2 enhancement results from left to right: (**a**) Original; (**b**) JED; (**c**) SEM; (**d**) Jiep; (**e**) SDD; (**f**) LUPP; (**g**) Our model.

In the enhancement results of Methods 1 [16] and 2 [15] in Figures 12 and 13, the visual impact of the images is low and the images are over-smoothed so that the image information is not effectively presented. In Figure 12, the visual effect of Method 5 [18] is close to that of the present paper, while the result of Method 4 [13] is slightly weaker than that of the present paper in terms of detail (The red rectangle box is shown in Figure 12). The overall brightness of Method 3 [14] is slightly fainter than that of Method 4 [13] and Method 5 [18] in this paper. In Figure 13, the results of Method 5 [18] show artifacts in some dark regions(The red rectangle box is shown in Figure 13), and the overall brightness of Method 3 [14] is slightly fainter than that of Method 4 [13] and Method 5 [18] in this paper.

## 5. Discussion

*5.1. Results Discussion*

By conducting a comparative analysis of the IQA values between the proposed method and other existing methods, it is evident that the average IQA values obtained from the proposed algorithm consistently outperform those obtained from the alternative methods. This observation holds true when considering both the dataset and real-world data. In summary, the IQA values derived from the proposed method exhibit superior performance across the board when compared to the other methods. In terms of subjective visual effects of images, the suggested algorithm effectively retains intricate image details while effectively reducing noise, surpassing alternative approaches in terms of subjective visual effects and boasting overall advantages.

*5.2. Limitation*

Despite being more comprehensive than alternative approaches, this paper's method does possess certain limitations. In the case of high noise intensity, the suppression effect of the adopted method is not obvious. The employed method demonstrates clear variability in its suppression effect when applied to different types of noise. For example, in the presence of speckle noise in low-illumination images, the images subjected to the proposed method still exhibit notable striping noise, despite the enhancement efforts. In the presence of Gaussian noise within the image, the suppression effect of the proposed method is evident. When salt-and-pepper noise is present in the image, the suppression effect of the proposed method is limited. To this end, it is necessary to establish a separate denoising model in the follow-up study.

## 6. Conclusions

This paper introduces a low-illumination image enhancement using local gradient relative deviation for Retinex model. From the perspective of texture and structure, the local gradient relative deviation is used as a constraint term, and the noise item is added to the model to highlight the texture and structure information and improve the robustness of the model. In addition, to achieve superior results in terms of smoothing the illumination component and preserving fine details of the reflectivity, this paper utilizes the $L_2 - L_P$ norm as a constraint on the model. Finally, this article adopts a standard optimization method to efficiently solve the optimization process. Extensive experiments demonstrate the superior performance of the proposed approach in comparison to other Retinex methods. Moreover, this approach can provide new ideas for further development of remote sensing image processing.

**Author Contributions:** Conceptualization, B.Y. and L.Z.; methodology, B.Y. and L.Z.; writing—original draft preparation, B.Y. and L.Z.; writing—review and editing, B.Y., X.W., L.Z., T.G. and X.C.; funding acquisition, L.Z. All authors have read and agreed to the published version of the manuscript.

**Funding:** This research was funded by the National Natural Science Foundation of China under Grant 62075219 and Grant 61805244; and the Key Technological Research Projects of Jilin Province, China under Grant 20190303094SF.

**Data Availability Statement:** The LOL data used in this paper are available at the following link: https://daooshee.github.io/BMVC2018website/, accessed on 15 February 2023. The AID data used in this paper are available at the following link: http://www.captain-whu.com/project/AID/, accessed on 15 February 2023. The VV data used in this paper are available at the following link: https://sites.google.com/site/vonikakis/datasets, accessed on 15 February 2023. The SCIE data used in this paper are available at the following link: https://github.com/csjcai/SICE, accessed on 15 February 2023. The TGRS-HRRSD data used in this paper are available at the following link: https://github.com/CrazyStoneonRoad/TGRS-HRRSD-Dataset/tree/master/OPT2017, accessed on 15 February 2023.

**Acknowledgments:** The authors would like to thank the anonymous reviewers for their valuable comments.

**Conflicts of Interest:** The authors declare no conflict of interest.

## Abbreviations

The following abbreviations are used in this manuscript:

| | |
|---|---|
| ADMM | Alternating Direction Multiplier Method |
| PCG | Preconditioned Conjugate Gradient |
| IQA | Image Quality Assessment |
| NIQE | Natural Image Quality Evaluator |
| CPCQI | Colorfulness-based PCQI |
| VIF | Visual Information Fidelity |
| ARISM | AutoRegressive-based Image Sharpness Metric |
| LOE | Lightness Order Error |

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
