# Peer review of "Low-Illumination Image Enhancement Using Local Gradient Relative Deviation for Retinex Models"

_remotesensing, doi:10.3390/rs15174327_

Round 1
Reviewer 1 Report
The proposed method of the article titled “low-illumination image enhancement using local gradient relative deviation for Retinex model” seems promising in addressing the challenges of low-illumination image enhancement and improving the overall visual quality while preserving the original information of the scene. However, by addressing the following points, the manuscript can be strengthened and provide a clearer understanding of the proposed method's performance and its advantages over existing techniques.
· By capitalizing the initial letters of each main word, the title becomes more visually balanced and easier to read.
· Clarify the specific values and ranges of the parameters α, β, γ, γ1, γ2, δ, ε. Providing more details about these parameters would help readers understand their significance and potential impact on the results.
· Provide a rationale or justification for the chosen values of the parameters, such as α, β, γ, γ1, γ2, δ. Explain why these specific values were selected and how they contribute to the effectiveness of the proposed model.
· Consider providing more information or references regarding the Preconditioned Conjugate Gradient (PCG) method used for rapid convergence. Please explain how this method improves computational efficiency and why it is suitable for the proposed algorithm.
· Clarify the selection of γ=2.2 for gamma correction in the illumination component correction step. Explain why this value was chosen and whether it was based on empirical observations or prior research.
· Instead of simply presenting the average values of the IQA metrics, briefly explain each metric and how it measures image quality. This will help readers better understand the significance of the results.
· If possible, perform a statistical significance analysis to determine if the differences between the proposed method and other techniques are statistically significant. This will strengthen the credibility of the results.
· While the figures show the enhanced results, providing a more detailed qualitative analysis would be helpful. Describe specific visual improvements, such as preserving fine details, reducing noise, or enhancing colors. This will give more insights into the strengths of the proposed method.
· If possible, increase the number of real image data samples used for evaluation. This will provide a more comprehensive assessment of the proposed method's performance in real-world scenarios.
· Minimize the use of personal pronouns such as "we," "our," or "I" to maintain an objective tone throughout the manuscript. Instead, focus on presenting the information and findings without emphasizing the researchers' involvement.
· When introducing abbreviations or acronyms, ensure that they are defined upon first use. This will help readers understand the meaning of these terms throughout the manuscript.
· The manuscript would benefit from overall improvements in English language usage, grammar, and sentence structure. Consider revising sentences for clarity and coherence.
Reviewer 2 Report
Nil
Author Response
Dear reviewer,
I would like to thank you for the time and effort spent on reviewing
the manuscript. Sincerely thank you again for your recognition of me and
the other authors, and for your recognition and support of our work.
Sincerely,
Reviewer 3 Report
The paper is well written and technically sound. The proposed solution outperforms in several aspects the state of the art, but it is very sensitive to noise. The equations could have been improved as well as mathematical notation. I would prefer the authors to use bold notation for matrices. This would clarify a lot of the equation derivation. Any lengthy derivation should be part of the annex and it should be very well referenced in the body of the document. The noisy regime doesn’t seem too difficult to address. The authors must try to show results with noise to see the effects on the proposed algorithm.
No comments
